# WHERE IS THE INFORMATION IN A DEEP NETWORK?

## ABSTRACT

Whatever information a deep neural network has gleaned from past data is en-coded in its weights. How this information affects the response of the network to future data is largely an open question. In fact, even how to define and mea-sure information in a network entails some subtleties. We measure information in the weights of a deep neural network as the optimal trade-off between accuracy of the network and complexity of the weights relative to a prior. Depending on the prior, the definition reduces to known information measures such as Shannon Mutual Information and Fisher Information, but in general it affords added flexi-bility that enables us to relate it to generalization, via the PAC-Bayes bound, and to invariance. For the latter, we introduce a notion of *effective information in the activations*, which are deterministic functions of future inputs. We relate this to the Information in the Weights, and use this result to show that models of low (in-formation) complexity not only generalize better, but are bound to learn invariant representations of future inputs. These relations hinge not only on the architecture of the model, but also on how it is trained.

## 1 INTRODUCTION

At the end of training a deep neural network, all that is left of past experience is a set of values stored in its weights. So, studying what "information" they contain seems like a natural starting point to understand how deep networks learn.

But how is the information in a deep neural network even defined? The weights are not a ran-dom variable, and the network outputs a deterministic function of its input, with degenerate (infi-nite) Shannon Mutual Information between the two. This presents a challenge for theories of Deep Learning based on Shannon Information (Saxe et al., 2018). Several frameworks have been devel-oped to reason about information in fixed sets of values, for instance by Fisher and Kolmogorov, but they either do not relate directly to relevant concepts in Deep Learning, such as generalization and invariance, or cannot be estimated in practice for modern deep neural networks (DNNs).

Beyond *how* they define information, existing theories of Deep Learning are limited by *whose* infor-mation they address: Most approaches focus on information of the *activations* of the network – the output of its layers – rather than their parameters, or weights, although recent information-theoretic approaches to study the weights are discussed in the next section. The weights are a representation *of* past data (the training set of inputs and outputs), trained *for* predicting statistics of the training set itself (*e.g.*, the output), *relative* to prior knowledge. The activations are a representation *of* (possibly unseen) future inputs (test set), ideally *sufficient* to predict future outputs, and *invariant* to nuisance variability in the data that should not affect the output. We have no access to future data, and the Shannon Information their representation contains does not account for the *finite* training set, hence missing a link to generalization.

But how are these properties of sufficiency and invariance achieved through the training process? Sufficiency alone is trivial — any invertible function of the data is, in theory, sufficient — but it comes at the expense of complexity[1] (or *minimality*) and invariance of the representation. Invariance alone is similarly trivial – any constant function is invariant. A learning criterion therefore must trade off accuracy, complexity and invariance. *The best achievable complexity trade-off is what we define*

---

[1] In this paper, we refer to complexity as *information* complexity, to be distinguished from complexity of the hypothesis space, for instance measured by the VC Dimension.

*as Information for the task*. The challenge is that we wish to characterize sufficiency and invariance of representations of the *test* data, while we only have access to the training set.

So, throughout this paper, we discuss four distinct concepts: (1) Sufficiency of the weights, captured by a training loss (*e.g.*, empirical cross-entropy); (2) complexity and minimality of the weights, captured by the information they contain; (3) sufficiency of the activations, captured by the test loss which we cannot compute, but can bound using the Information in the Weights; (4) invariance of the activations, a property of the test data, which is not explicitly present in the formulation of the learning process when training a deep neural network. To do all that, we first need to formally define both *information of the weights* and of the *activations*.

## 1.1 SUMMARY OF CONTRIBUTIONS AND RELATED WORK

Our first contribution is to measure the Information in the Weights of a deep neural network as the trade-off between the amount of noise we *could* add to the weights (measured by its entropy relative to a prior), and the performance the network would achieve in the task at hand. Informally, given an encoding algorithm, this is the number of bits needed to encode the weights in order to solve the task at some level of precision, as customary in Rate-Distortion Theory. The optimal trade-off traces a curve that depends on the task and the architecture, and solutions along the curve can be found by optimizing an *Information Lagrangian*. The Information Lagrangian is in the general form of an Information Bottleneck (IB) (Tishby et al., 1999), but is fundamentally different from the IB used in most prior work in deep learning (Tishby & Zaslavsky, 2015), which refers to the activations, rather than the weights. Our measure of information is practical even in large-scale networks, with millions of parameters, and retains the dependency on the number of samples in the training set.

Our second contribution is to derive a relation between the two informations (Section 4), where we show that the Information Lagrangian of the weights of deep networks bounds the Information Bottleneck of the activations, but not vice-versa. This is important, as the IB of the activations is degenerate when computed on the training set, hence cannot be used at training time to enforce properties. On the other hand, the Information Lagrangian of the weights remains well defined, and through our bound it controls invariance at test time.

Our method requires specifying a parametrized noise distribution, as well as a prior, to measure information. While this may seem undesirable, we believe it is essential and key to the flexibility of the method, as it allows us to compute concrete quantities, tailored to DNNs, that relate generalization and invariance in novel ways. Of all possible choices of noise and prior to compute the Information in the Weights, there are a few standard ones: An *uninformative* prior yields the Fisher Information of the weights. A prior obtained by averaging training over all relevant datasets yields the Shannon mutual information between the dataset (now a random variable) and the weights. A third important choice is the noise distribution induced by stochastic gradient descent (SGD) during the training process, which we discuss in the paper.

As it turns out, all three resulting notions of information are important to understand learning in deep networks: Shannon's relates closely to generalization, via the PAC-Bayes Bound (Section 3.1). Fisher's relates closely to invariance in the representation of test data (activations) as we show in Section 4. The noise distribution of SGD is what connects the two, and establishes the link between invariance and generalization. Although it is possible to minimize Fisher or Shannon Information independently, we show that when the weights are learned using SGD, the two are related. This is our third contribution, which is made possible by the flexibility of our framework (Section 3.3). Finally, in Section 5 we discuss open problems and further relations with prior work.

There is a growing literature on information and generalization bounds for the weights of deep networks (Xu & Raginsky, 2017; Pensia et al., 2018). Given a data generating distribution $\mathcal{D} \sim \mu(x, y)$, a training algorithm $w = A(\mathcal{D})$ is said to be $(\epsilon, \mu)$-information stable if $I(w; \mathcal{D}) < \epsilon$. The generalization gap of a training algorithm can then be bounded in terms of its information stability. Indeed, the quantity $I(w; \mathcal{D})$ is related to our definition of information in the weights (Section 3.2), but we emphasize that our general definition of amount of information in the weights extends to the case where both the dataset $\mathcal{D}$ and $\mathbf{w}$ are assumed to be given and fixed (as it is often common in Deep Learning), and not resampled every time. First, some of our main results are to prove that convergence to flat minima (low Fisher Information) and "path" stability of SGD (Hardt et al., 2015) imply "information" stability in the sense of Xu & Raginsky (2017) (Proposition 3.7). Unlike those

works, our bound does not depend solely on the noise induced by the steps of SGD, but also the geometry of the loss lanscape, which allows to better capture some fundamental properties of Deep Learning. Second, while those work only bound the generalization performance on the training task, we connect the information stability with amount of information in the activations of a DNN and invariance to nuisances (Section 4). This this can be used to guarantee the quality of the learned representation in a transfer learning setting.

Note that the fact that some weights can be perturbed at little loss has been known empirically for a while (Hinton & Van Camp, 1993). We exploit this property to define information for a particular set of weights, in a manner that is quite distinct from standard PAC-Bayes, using Fisher's information instead.

This paper takes Achille et al. (2019) as the starting point of investigation, attempting to measure the quantities at play more accurately, which leads us beyond Shannon's formalism, to a more general setting that also includes Fisher's formalism and the relation between the two, mediated by the properties of deep neural networks and SGD. All the (information) quantities we measure are specific to a particular weight vector, not its distribution.

## 2  PRELIMINARIES AND NOTATION

We denote with $x \in X$ an input (*e.g.*, an image), and with $y \in Y$ a "task variable," a random variable which we are trying to infer, *e.g.*, a label $Y = \{1, \ldots, C\}$. A dataset is a finite collection of samples $\mathcal{D} = \{(x_i, y_i)\}_{i=1}^N$ that *specify* the task. A DNN model trained with the cross-entropy loss encodes a conditional distribution $p_w(y|x)$, parametrized by the weights $w$, meant to approximate the posterior of the task variable $y$ given the input $x$. The Kullbach-Liebler, or *KL-divergence,* is the relative entropy between $p(x)$ and $q(x)$: $\mathrm{KL}(\, p(x) \,\|\, q(x) \,) = \mathbb{E}_{x \sim p(x)}\big[\log(p(x)/q(x))\big]$. It is always non-negative, and zero if and only if $p(x) = q(x)$. It measures the (asymmetric) similarity between two distributions. Given a family of conditional distributions $p_w(y|x)$ parametrized by a vector $w$, we can ask how much perturbing the parameter $w$ by a small amount $\delta w$ will change the distribution, as measured by the KL-divergence. To second-order, this is given by $\mathbb{E}_x \mathrm{KL}(\, p_w(y|x) \,\|\, p_{w+\delta w}(y|x) \,) = \delta w^t F \delta w + o(\|\delta w\|^2)$ where $F$ is the *Fisher Information Matrix* (or simply "Fisher"), defined by $F = \mathbb{E}_{x,y \sim p(x) p_w(y|x)}[\nabla \log p_w(y|x)^t \nabla \log p_w(y|x)] = \mathbb{E}_{x \sim p(x) p_w(y|x)}[-\nabla_w^2 \log p_w(y|x)]$. For its relevant properties see Martens (2014). It is important to notice that the Fisher depends on the ground-truth data distribution $p(x, y)$ only through the domain variable $x$, not the *task variable* $y$, since $y \sim p_w(y|x)$ is sampled from the model distribution when computing the Fisher. This property will be used later.

Given two random variables $x$ and $z$, their *Shannon mutual information* is defined as $I(x; z) = \mathbb{E}_{x \sim p(x)}[\mathrm{KL}(\, p(z|x) \,\|\, p(z) \,)]$ that is, the expected divergence between the distribution of $z$ after an observation of $x$, and the prior distribution of $z$. It is positive, symmetric, zero if and only if the variables are independent (Cover & Thomas, 2012), and measured in Nats when using the natural logarithm.

In supervised classification one is usually interested in finding weights $w$ that minimize the cross-entropy loss $L_{\mathcal{D}}(w) = \mathbb{E}_{(x,y) \sim \mathcal{D}}[-\log p_w(y|x)]$ on the training set $\mathcal{D}$. The loss $L_{\mathcal{D}}(w)$ is usually minimized using *stochastic gradient descent* (SGD), which updates the weights $w$ with an estimate of the gradient computed from a small number of samples (mini-batch). That is, $w_{k+1} = w_k - \eta \nabla \hat{L}_{\xi_k}(w)$, where $\xi_k$ are the indices of a randomly sampled mini-batch and $\hat{L}_{\xi_k}(w) = \frac{1}{|\xi_k|} \sum_{i \in \xi_k} [-\log p_w(y_i|x_i)]$. Notice that $\mathbb{E}_{\xi_k}[\nabla \hat{L}_{\xi_k}(w)] = \nabla L_{\mathcal{D}}(w)$, so we can think of the mini-batch gradient $\nabla \hat{L}_{\xi_k}(w)$ as a noisy version of the real gradient. Using this intuition we can write:

$$w_{k+1} = w_k - \eta \nabla L_{\mathcal{D}}(w_k) + \sqrt{\eta}\, T_{\xi_k}(w_k) \tag{1}$$

with the induced "noise" term $T_{\epsilon_k}(w) = \sqrt{\eta}\,\big(\nabla \hat{L}_{\xi_k}(w) - \nabla L(w)\big)$. Written in this form, eq. (1) is a Langevin diffusion process, with (non-isotropic) noise $T_{\xi_k}$ Li et al. (2017); Chaudhari & Soatto (2018).

## 3 INFORMATION IN THE WEIGHTS

One could define the Information in the Weights as their coding length after training. This, however, would not be meaningful, as only a small subset of the weights matters: If perturbing a given weight configuration $w$ ($w' \leftarrow w + \delta w$) were to yield no change in the cross-entropy loss (*i.e.*, $L_\mathcal{D}(w') \approx L_\mathcal{D}(w)$), one could argue that such weights contain "no information" about the task. Storing those weights with low precision, or pruning them, or randomizing them, would yield no performance loss. On the other hand, if slightly perturbing a configuration of weights were to yield a large increase in the loss, one could argue that such weights are very "informative," and store them with high precision. But what perturbations should one consider (*e.g.*, additive or multiplicative)? And how "small" should they be? What distribution should the perturbations be drawn from? To address these issues, we introduce the following definition:

**Definition 3.1** (Information in the Weight). *The complexity of the task $\mathcal{D}$ at level $\beta$, using the post-distribution $Q(w|\mathcal{D})$ and the pre-distribution $P(w)$, is*

$$C_\beta(\mathcal{D}; P, Q) = \mathbb{E}_{w \sim Q(w|\mathcal{D})}[L_\mathcal{D}(p_w(y|x))] + \beta \underbrace{\mathrm{KL}(\, Q(w|\mathcal{D}) \,\|\, P(w) \,)}_{\text{Information in the Weights}}, \tag{2}$$

*where $\mathbb{E}_{w \sim Q(w|\mathcal{D})}[L_\mathcal{D}(p_w(y|x))]$ is the reconstruction error under the distribution $Q(w|\mathcal{D})$; $\mathrm{KL}(\, Q(w|\mathcal{D}) \,\|\, P(w) \,)$ measures the entropy of $Q(w|\mathcal{D})$ relative to the $P(w)$. If $Q^*(w|\mathcal{D})$ minimizes eq.* (2) *for a given $\beta$, we call $\mathrm{KL}(\, Q^*(w|\mathcal{D}) \,\|\, P(w) \,)$ the amount of Information in the Weights for the task $\mathcal{D}$ at level $\beta$ (or "Information in the Weights," when the meaning is clear from the context).*

Note that the definition of information is based on the loss $L_\mathcal{D}$ on the training set, which depends on the number of samples in $\mathcal{D}$, and does not require access to the underlying data distribution. We call $Q(w|\mathcal{D})$ a "post-distribution" because it is picked after seeing the dataset $\mathcal{D}$. We do not call it "posterior" to emphasize that it is an arbitrary distribution and does not have any Bayesian interpretation. Similarly, $P(w)$ is an arbitrary "pre-distribution", distinct from a Bayesian "prior," picked before the dataset is seen.[2] Having stressed the arbitrary nature of $P(w)$ and $Q(w|\mathcal{D})$, from now on with simply call them prior and posterior for simplicity. We also refer to the variability implied by $Q$ as "noise" even though the mechanism by which it acts could be deterministic.

Special cases of (2) include the case $\beta = 1$, when eq. (2) formally coincides with the evidence lower-bound (ELBO) used to train Bayesian Neural Networks. However, while the ELBO assumes the existence of a Bayesian posterior $P(w|\mathcal{D})$ of which $Q(w|\mathcal{D})$ is an approximation, we require no such assumption. The use of the ELBO for different values of $\beta$ and the connection with rate-distortion theory has also been explored in Hu et al. (2018). Closer to our viewpoint is Hinton & Van Camp (1993), that shows that, for $\beta = 1$, eq. (2) is the cost to encode the labels in $\mathcal{D}$ together with the weights of the network. This justifies considering, for any choice of $P$ and $Q$, the term $\mathrm{KL}(\, Q^*(w|\mathcal{D}) \,\|\, P(w) \,)$ as the coding length of the weights using some algorithm, although this is true only if they are encoded together with the dataset. A drawback with these approaches is that they lead to non-trivial results only if $\mathrm{KL}(\, Q(w|\mathcal{D} \,\|\, P(w) \,)$ is much smaller than the coding length of the labels in the dataset (*i.e.*, $N \log |Y|$ nats, assuming a uniform label distribution). Unfortunately, this is far from being the case with typical deep neural networks.

Rather than the particular value of the coding length, we focus on *how it changes as a function of $\beta$* for a given noise model, tracing a Pareto-optimal curve which defines the Information in the Weights we have proposed above.

### 3.1 INFORMATION IN THE WEIGHTS CONTROLS GENERALIZATION

Equation (2) defines a notion of information that, while related to the learning task, does not immediately relate to generalization error or invariance of the representation. Throughout the rest of this work, we build such connections leveraging on existing work. We start by using the well-known PAC-Bayes bound (McAllester, 2013) to connect the information that the weights retain about the training set to performance on the test data.

---

[2]Note that this definition is compatible with a deterministic training process, or with a stochastic training process that yields a point estimate of the weights, or with a stochastic training process that yields a distribution of weights.

**Theorem 3.2** (McAllester (2013), Theorems 2-4). *Assume the dataset $\mathcal{D} = \{(x_i, y_i)\}_{i=1}^N$ is sampled i.i.d. from a distribution $p(y, x)$, and assume that the per-sample loss used for training is bounded by $L_{max} = 1$ (we can reduce to this case by clipping and rescaling the loss). For any fixed $\beta > 1/2$, prior $P(w)$, and weight distribution $Q(w|\mathcal{D})$, with probability at least $1 - \delta$ over the sample of $\mathcal{D}$, we have:*

$$L_{test}(Q) \leq \frac{1}{N(1 - \frac{1}{2\beta})}\Big[\mathbb{E}_{w \sim Q(w|\mathcal{D})}[L_{\mathcal{D}}(p_w)] + \beta\, \mathrm{KL}(\, Q \,\|\, P\,) + \beta \log \frac{1}{\delta}\Big], \qquad (3)$$

*where $L_{test}(Q) := \mathbb{E}_{x,y \sim p(x,y)}[\mathbb{E}_{w \sim Q}[p_w(y|x)]]$ is the expected per-sample test error that the model incurs using the weight distribution $Q(w|\mathcal{D})$. Moreover, given a distribution $p(\mathcal{D})$ over the datasets, we have the following bound in expectation over all possible datasets:*

$$\mathbb{E}_{\mathcal{D}}[L_{test}(Q(w|\mathcal{D}))] \leq \frac{1}{N(1 - \frac{1}{2\beta})}\Big[\mathbb{E}_{\mathcal{D}}[L_{\mathcal{D}}(Q(w|\mathcal{D}))] + \beta\, \mathbb{E}_{\mathcal{D}}[\mathrm{KL}(\, Q(w|\mathcal{D}) \,\|\, P\,)]\Big]. \qquad (4)$$

Hence, minimizing the complexity $C_\beta(\mathcal{D}; P, Q)$ can be interpreted as minimizing an upper-bound on the test error, rather than merely minimizing the training error. In Dziugaite & Roy (2017), a non-vacuous generalization bound is computed for DNNs, using a (non-centered and non-isotropic) Gaussian prior and Gaussian posterior distributions.

## 3.2 Shannon vs. Fisher Information in the Weights

Definition 3.1 depends on an arbitrary choice of the noise distribution and of the prior. While this may appear cumbersome, it captures the fact that to properly measure the information in a deep network we need to *adapt the choice of noise to the model.* In this section, we show how different priors and posteriors result in known definitions of information, in particular Shannon's and Fisher's. This section is inspired by Achille et al. (2019), who derive these relations in the even more general setting of Kolmogorov Complexity.

In some cases, there may be an actual distribution $\pi(\mathcal{D})$ over the possible training sets, so we may aim to find the prior $P(w)$ that minimizes the expected test error bound in eq. (4), which we call *adapted prior.* The following proposition shows that the information measure that minimizes the bound in expectation is the Shannon Mutual Information between weights and dataset.

**Proposition 3.3** (Shannon Information in the Weights). *Assume the dataset $\mathcal{D}$ is sampled from a distribution $\pi(\mathcal{D})$, and let the outcome of training on a sampled dataset $\mathcal{D}$ be described by a distribution $Q(w|\mathcal{D})$. Then the prior $P(w)$ minimizing the expected complexity $\mathbb{E}_{\mathcal{D}}[C_\beta(\mathcal{D}; P, Q)]$ is the marginal $P(w) = \mathbb{E}_{\mathcal{D}}[Q(w|\mathcal{D})]$, and the expected Information in the Weights is given by*

$$\mathbb{E}_{\mathcal{D}}[\mathrm{KL}(\, Q(w|\mathcal{D}) \,\|\, P(w)\,)] = I(w; \mathcal{D}). \qquad (5)$$

*Here $I(w; \mathcal{D})$ is Shannon's mutual information between the weights and the dataset, where the weights are seen as a (stochastic) function of the dataset given by the training algorithm (SGD).*

The above proposition is textbook material, but it is proven in the Appendix for completeness. Note that, in this case, the prior $P(w)$ is optimal given the choice of the training algorithm (*i.e.*, the map $A : \mathcal{D} \to Q(w|\mathcal{D})$) and the distribution of training datasets $\pi(\mathcal{D})$. Using this prior we have the following expression for the expectation over $\mathcal{D}$ of eq. (2):

$$\mathbb{E}_{\mathcal{D}}[C_\beta(\mathcal{D}; P, Q)] = \mathbb{E}_{\mathcal{D}}[\mathbb{E}_{w \sim Q(w|\mathcal{D})}[L_{\mathcal{D}}(w)]] + \beta I(w; \mathcal{D}). \qquad (6)$$

Notice that this is the general form of an Information Bottleneck (Tishby et al., 1999). However, the use of the IB in Deep Learning has focused on the activations (Shwartz-Ziv & Tishby, 2017), which are the bottleneck between the inputs $x$ and the output $y$. Instead, the **Information Lagrangian** eq. (6) concerns the weights of the network, which are the bottleneck between the training dataset $\mathcal{D}$ and inference on the future test distribution. Hence, it directly relates to the training process, the finite nature of the dataset, and can yield bounds on future performance. The particular Information Bottleneck for the weights was first used by Achille & Soatto (2018), but derived in a more limited setting that did not allow the flexibility needed to establish the bounds we describe in this paper using Fisher Information.

While the adapted prior of Proposition 3.3 allows us to compute an optimal generalization bound, it requires averaging with respect to all possible datasets, which requires knowledge of the task

distribution $\pi(\mathcal{D})$ and is, in general, unrealistic for deep learning. At the other extreme, we can consider an uninformative prior, and obtain the (log-determinant of the) Fisher as a measure of information.

**Proposition 3.4** (Fisher Information in the Weights). *Assume an isotropic Gaussian prior $P(w) \sim N(0, \lambda^2 I)$ and a Gaussian posterior $Q(w|\mathcal{D}) \sim N(w^*, \Sigma)$, where $w^*$ is any global minimizer of the cross-entropy loss. Then, for $\lambda \to \infty$, as the prior becomes (improper and) uninformative, we have that:*

1. *For small $\beta$, the covariance $\Sigma^*$ that minimizes $C_\beta(\mathcal{D}; P, Q)$ tends to $\frac{\beta}{2} H^{-1} = \frac{\beta}{2} F^{-1}$, in accordance with the Cramér-Rao bound, where $H = \nabla_w^2 L_\mathcal{D}(w)$ is the Hessian of the cross-entropy loss, and $F$ is the Fisher Information Matrix;*

2. *The Information in the Weights is given by*

$$\mathrm{KL}(\, Q(w|\mathcal{D}) \, \| \, P(w) \,) = \frac{1}{2} \log |F| + \frac{1}{2} k \log \lambda^2 + O(1). \tag{7}$$

*Note that the constant $\frac{1}{2} k \log \lambda^2$ does not depend on $Q(w|\mathcal{D})$ and hence can be ignored.*

**Remark 3.5.** The above proposition assumes that the configuration of the weights to which we are adding noise is a global minimum, in which case the Hessian and the Fisher matrix coincide. In fact, we have the following decomposition of the Hessian (Martens, 2014, eq. 6 and Sect. 9.2):

$$H = F + \frac{1}{N} \sum_{(x_i, y_i) \in \mathcal{D}} \sum_{j=1}^{k} \left[ \nabla_z L(y_i, z)|_{z=f_w(x_i)} \right]_j H_{[f]_j}, \tag{8}$$

where $z = f_w(x_i)$ is the output of the network for input $x_i$, $L(y_i, z) = -\sum_{j=1}^{k} \delta_{y_i, j} \log(z_j)$ is the cross-entropy loss for the $i$-th sample, and $H_{[f]_j}$ is the Hessian of the $j$-th component of $z$. If most training samples are predicted correctly, then $\nabla_z L(y_i, z) \approx 0$ and $H \approx F$. Otherwise, there is no guarantee that $H$ will be positive definite, making the second-order approximation used in Proposition 3.4 invalid, since it suggests that adding noise along the negative directions can decrease the loss unboundedly. Following Martens (2014), we use a more robust second-order approximation by ignoring the second part of eq. (8), hence using the Fisher as a stable positive semi-definite approximation of the curvature. In this setting, eq. (7) remains valid at all points. The additive term diverging to infinity is expected when using an improper prior, and does not affect considerations about the gradients or the minimizers. Note that there is no assumption that the curvature of the loss be constant near convergence.

## 3.3 Information in the Learning Dynamics

In Section 3.1 we have seen that the Shannon Information of the weights controls generalization. In Section 4 we will see that the Fisher controls invariance of the activations. Can we just pick one measure of information and use it to characterize both generalization and invariance?

In principle, the Fisher can also be used in Theorem 3.2 to obtain generalization bounds; however, it is likely to give a vacuous bound if used directly, as it is usually much larger than the optimal Shannon Information. In this section, we argue that, for a deep network trained with stochastic gradient descent *on a given domain,* Fisher and Shannon go hand-in-hand. This hinges on the fact that: (i) The Fisher depends on the domain, but not on the labels, hence all tasks sharing the same domain share the same Fisher, (ii) SGD implicitly minimizes the Fisher, hence, (iii) SGD tends to concentrate the solutions in a restricted area of low Fisher solutions, hence minimizing the Shannon Information.

While (i) follows directly from the definition of the Fisher Information Matrix (Section 2), (ii) is not immediate, as SGD does not explicitly minimize the Fisher. The result hinges on the fact that, by adding noise to the optimization process, SGD will tend to escape sharp minima, and hence, since the Fisher is a measure of the curvature of the loss function, it will evade solutions with high Fisher. We can formalize this reasoning using a slight reformulation of the Eyring–Kramers law (Berglund, 2011) for stochastic processes in the form of eq. (1).

**Proposition 3.6** (Berglund (2011), eq. 1.9). *Let $w^*$ be a local minimizer of the loss function $L_\mathcal{D}(w^*)$. Consider the path $\gamma$ joining $w^*$ with any other minimum which has the least increase*

*in the loss function. The point with the highest loss along the path is a saddle point $w^s$ (the* rele-*vant saddle) with a single negative eigenvalue $\lambda_1(w^s)$. Then, in the limit of small step size $\eta$, and assuming isotropic gradient noise, the expected time before SGD escapes the minimum $w^*$ is given by*

$$\mathbb{E}[\tau] = \frac{2\pi}{|\lambda_1(w^s)|} e^{\frac{1}{T}(\mathcal{F}(w^s) - \mathcal{F}(w^*))},$$

*where we have defined the* free energy $\mathcal{F}(w) = L_{\mathcal{D}}(w) + \frac{T}{2} \log |F(w)|$*, where $F(w)$ is the Fisher computed at $w$, and $T \propto \eta/B$, where $B$ is the batch size. In particular, increasing $T$ (the "temperature" of SGD) makes SGD more likely to avoid minima with high Fisher Information.*

We can informally summarize the above statement as saying that SGD, rather than minimizing directly the loss function, minimizes a *free energy* $\mathcal{F}(w) = L_{\mathcal{D}}(w) + \frac{T}{2} \log |F(w)|$. Hence, the Fisher Information in the Weights *controls the dynamics* by slowing down learning when more information needs to be stored, as made precise in the next proposition. Note that the improper nature of the prior plays no role in the free energy (constants do not matter); the slowing down corresponding to large Fisher information has also been observed empirically by Achille et al. (2018).

We can now finally prove (iii), connecting the Fisher Information with the Shannon, which are at face value unrelated. The proof leverages an approximation of the mutual information using the Fisher Information presented in Brunel & Nadal (1998).

**Proposition 3.7.** *Assume the space of datasets $\mathcal{D}$ admits a differentiable parametrization.[3] Assume that $p(\mathcal{D}|w)$ is concentrated along a single dataset (i.e., the one used for training). Then, we have the approximation:*

$$I(w; \mathcal{D}) \approx H(\mathcal{D}) - \mathbb{E}_{\mathcal{D}} \left[ \frac{1}{2} \log \left( \frac{(2\pi e)^k}{|\nabla_{\mathcal{D}} w^{*t} F_w(w^*) \nabla_{\mathcal{D}} w^*|} \right) \right]$$

*where $H$ is the entropy and we assume $p(w|\mathcal{D}) = N(w^*(\mathcal{D}), F(w^*(\mathcal{D}))^{-1})$; $w^*(\mathcal{D})$ are the weights obtained at the end of training on dataset $\mathcal{D}$, and we assume that $\partial_{\mathcal{D}} F(w^*(\mathcal{D})) \ll \nabla_{\mathcal{D}} w^*(\mathcal{D})$.[4] The term $\nabla_{\mathcal{D}} w^*$ is the Jacobian of the final point with respect to changes of the training set.*

Notice that the norm $\|\nabla_{\mathcal{D}} w^*\|$ of the Jacobian $\nabla_{\mathcal{D}} w^*$ can be interpreted as a measure of the stability of SGD, that is, how much the final solution changes if the dataset is perturbed (Hardt et al., 2015). Hence, reducing the Fisher $F_w(w^*(\mathcal{D}))$ of the final weights found by SGD (*i.e.*, the flatness of the minimum), or making SGD more stable, *i.e.*, reducing $\nabla_{\mathcal{D}} w^*(\mathcal{D})$, both reduce the mutual information $I(w; \mathcal{D})$, and hence improve generalization per the PAC-Bayes bound. The Gaussian assumption for the weight distribution is for convenience of computation.

## 4 THE ROLE OF INFORMATION IN THE INVARIANCE OF THE REPRESENTATION

Thus far we have seen that training a DNN using SGD recovers weights that are a sufficient ($w$ minimizes the training loss) and minimal (they have low Information, either Shannon's or Fisher's) representation of the training dataset $\mathcal{D}$. The PAC-Bayes Bound guarantees that, on average, sufficiency of weights – a representation of the training set – implies sufficiency of the activations – a representation of the input datum at test time. What we are missing is a guarantee that, in addition to being sufficient, the representation of the test datum is also minimal, that is the information in the *activations* is also minimized. Why do we care that the representation of future data be minimal? Because it has been established by Achille & Soatto (2018) that this leads to invariance of the representation to nuisance variability (a sufficient representation is invariant if and only if it is minimal). In this section, we derive this missing link. In this case, minimality should be expressed in terms of Shannon's Mutual Information, since we want the representation to be invariant on average over future data, to which we do not have access (unlike the training dataset, which was the subject of previous sections). This entails some subtleties that have caused some confusion in the literature (Saxe

---

[3]For example by parametrizing the labels by the weights of an overfitting model, and sampling through a differentiable sampling algorithm.

[4]That is, that the Fisher does not change much if we perturb the dataset slightly. This assumption is mainly to keep the expression uncluttered, and a similar result can be derived without this additional hypothesis.

et al., 2018; Chelombiev et al., 2019). After the training is complete, activations are a deterministic function of the input, so some information-theoretic quantities are degenerate. Achille & Soatto (2018) argue that the weights of a DNN should be considered stochastic, where the stochasticity is imputed by the amount of information they store, and prove that the the mutual information between activations and inputs is in fact upper-bounded by Information in the Weights, considered as a noisy communication channel. A similar point of view was taken later by Goldfeld et al. (2018), who estimate mutual information under the hypothesis of inputs with isotropic noise. Both Achille & Soatto (2018) and Shwartz-Ziv & Tishby (2017) suggest connecting the noise in the weights and/or activations with the noise of SGD, although no formal connection has been established thus far.

Our main contribution, developed in this section, is to establish the connection between minimality of the weights and invariance of the activations, which resolves conflicting points of view. First, for a fixed deterministic DNNs, without stochasticity, we introduce the notion of *effective information* in the activations which, rather than measuring the information that an optimal decoder could extract from the activations, measures the information that the network effectively uses in order to classify. Using this definition, we show that *the Fisher Information in the Weights bounds both the Fisher and Shannon Information in the activations.* Notice that we already related the Fisher Information to the noise of SGD in Proposition 3.6.

### 4.1 Induced Stochasticity and Effective Information in the Activations

We denote with $z = f_w(x)$ the activations of a generic intermediate layer of a DNN, a deterministic function of $x$. According to the definition of Information in the Weights, small perturbations of uninformative weights cause small perturbations in the loss. Hence, information in the activations that is not preserved by such perturbations is not used by the classifier. This suggests the following definition.

**Definition 4.1.** *(Effective Information in the Activations) Let $w$ be the value of the weights, and let $n \sim N(0, \Sigma_w^*)$, with $\Sigma_w^* = \beta F^{-1}(w)$ be the Gaussian noise minimizing eq. (2) at level $\beta$ for an uninformative prior (Proposition 3.4). We call* effective information *(at noise level $\beta$) the amount of information about $x$ that is not destroyed by the added noise:*

$$I_{eff,\beta}(x;z) = I(x;z_n), \tag{9}$$

*where $z_n = f_{w+n}(x)$ are the activations computed by the perturbed weights $w + n \sim N(w, \Sigma_w^*)$.*

Using this definition, we obtain the following characterization of the information in the activations.

**Proposition 4.2.** *For small values of $\beta$ we have:*

*(i) The Fisher Information $F_{z|x} = \mathbb{E}_z[\nabla_x^2 \log p(z|x)]$ of the activations w.r.t. the inputs is:*

$$F_{z|x} = \frac{1}{\beta} \nabla_x f_w \cdot J_f F_w J_f^t \, \nabla_x f_w,$$

*where $\nabla_x f_w(x)$ is the Jacobian of the representation given the input, and $J_f(x)$ is the Jacobian of the representation with respect to the weights. In particular, the Fisher of the activations goes to zero when the Fisher of the weights $F_w$ goes to zero.*

*(ii) Under the hypothesis that, for any representation $z$, the distribution $p(x|z)$ of inputs that could generate it concentrates around its maximum, we have:*

$$I_{eff,\beta}(x;z) \approx H(x) - \mathbb{E}_x\left[\frac{1}{2}\log\left(\frac{(2\pi e)^k}{|F_{z|x}|}\right)\right], \tag{10}$$

*hence, by the previous point, when the Fisher Information of the weights decreases, the effective mutual information between inputs and activations also decreases.*

Hence, decreasing the Fisher Information that the weights have about the training set (which can be done by increasing the noise of SGD) decreases the apparently unrelated effective information between inputs and activations at test time. Moreover, making $\nabla_x f_w(x)$ small, *i.e.*, reducing the *Lipschitz constant* of the network, also reduces the effective information.

We say that a random variable $n$ is a **nuisance** for the task $y$ if $n$ affects the input $x$ but is not informative of $y$, *i.e.* $I(n, y) = 0$. We say that a representation $z$ is maximally invariant to $n$ if

$I(z, n)$ is minimal among all sufficient representations, which are all[5] the representations $z$ that capture all the information about the task contained in the input, $I(z, y) = I(x, y)$. The following claim in Achille & Soatto (2018), connects invariance to compression (minimality):

**Proposition 4.3** (Achille & Soatto (2018), Proposition 3.1). *A representation $z$ is maximally invariant to all nuisances at the same time if and only if $I(x, z)$ is minimal among the sufficient representations.*

Together with Proposition 4.2, this shows that a network which has minimal complexity (*i.e.*, minimal Information in the Weights) is forced to learn a representation that is *effectively invariant* to nuisances; that is, invariance emerge naturally during training by reducing the amount of information stored in the weights.

As a side note, we may wonder what distribution of the inputs would maximize the effective mutual information $I_{\text{eff}, \beta}(x; z)$; that is, what distribution the network is maximally adapted to represent (Brunel & Nadal, 1998). Maximizing $I_{\text{eff}, \beta}(x; z)$ with respect to $p(x)$ we obtain: $p^*(x) = \sqrt{|F_{z|x}|} / \int \sqrt{|F_{z|x}|} dx$. Using this, we obtain the following bound on the mutual information $I_{\text{eff}, \beta}(x; z)$ *for any input distribution*:

$$I_{\text{eff}}(x; z) \leq \log \left( \int \sqrt{|F_{z|x}|} dx \right).$$

Intuitively, this can be interpreted as the volume of the representation space; that is, how many well separated representations $z$ can be obtained mapping inputs $x$, taking into account that, because of a small Lipschitz constant of the network, or because of noise, multiple inputs may be mapped to similar representations.

## 5 DISCUSSION

Once trained, deep neural networks are deterministic functions of their input, and we are interested in understanding what "information" they retain, what they discard, and how they process unseen data. Ideally, we would like them to process future data by retaining all that matters for the task (sufficiency) and discarding all that does not (nuisance variability), leading to invariance. But we do not access to the test data, and the literature does not provide a rigorous or even formal connection between properties of the training set and invariance to nuisance variability in the test data.

This paper extends and develops results of Achille & Soatto (2018) and is, to the best of our knowledge, the first to define the information in a deep network, which is in the *weights* that represent the training set, in a way that connects it to generalization and invariance, which are properties of the activations of the test data. This Information in the Weights is neither Shannon's (used in Achille & Soatto (2018)) nor Fisher's, but a more general one that encompasses the two as special cases.

We leverage several existing results in the literature: the Information Lagrangian is introduced in Achille & Soatto (2018), but we extend it beyond Shannon Information, which presents some challenges when the source of stochasticity is not explicit. We draw on Fisher's Information, that formalizes a notion of *sensitivity* of a set of parameters, and is not tied to a particular assumption of generative model. We leverage the PAC-Bayes bound to connect sufficiency of the weights to sufficiency of the activations, and provide the critical missing link to connect minimality of the weights – that arises from the inductive bias of SGD when training deep networks – with minimality of the activations.

We put the emphasis on the distinction between Information in the Weights, as done by Achille & Soatto (2018), and information in the activations, which several other information-theoretic approaches to Deep Learning have focused. One pertains to representations of past data, which we can measure. The other pertains to desirable properties of future data, that we cannot measure, but we can bound. We provide a measurable bound, exploiting the Fisher Information, which enables reasoning about "effective stochasticity" even if a network is a deterministic function.

Our results connect to generalization bounds through PAC-Bayes, and account for the finite nature of the training set, unlike several other information-theoretic approaches to Deep Learning that only provide results in expectation.

---

[5]There are infinitely many, for instance the trivial function $z = x$, and any invertible function of it.

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

# A   EMPIRICAL VALIDATION

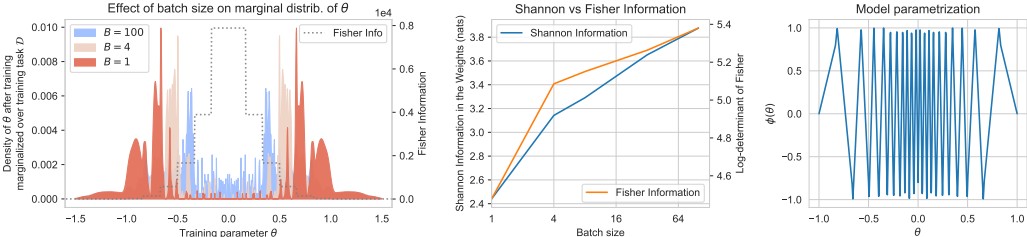

Figure 1: **(Left)** Plot of the marginal distribution of the parameter $w$ at the end of training, marginalized over all possible training tasks $\mathcal{D}$, as the batch size $B$ of SGD changes. Notice that as the batch size gets smaller SGD shifts farther away from areas with high Fisher Information (dotted line), supporting Proposition 3.6. **(Center)** Effect of the batch size on the Information in the Weights. Remarkably, while changing the batch size should only affect the Fisher Information, it also reduces the Shannon Information of the weights following the same qualitative dependence, in support of Proposition 3.7. **(Right)** Redundant parametrization $\phi(\theta)$ used in the experiment to emulate some of the key properties of the loss landscape of deep networks.

**Relation between Fisher and Shannon Information**    In this section we want to empirically verify the link between Fisher Information and Shannon Information of in a machine learning problem trained with SGD, which we explored in Section 3.3. Our main objective is to verify that decreasing the Fisher Information (which can be done by changing the hyper-parameters of SGD, in particular the batch size and learning rate (Proposition 3.6), does indeed decrease the Shannon Information (Figure 1, center). It is known in general that the Fisher Information can be used to upper-bound the Shannon Information (recall from Section 3.2 that Shannon Information is the minimum attainable). However, we show empirically here that even for a simple 1D example this bound is remarkably loose (by orders of magnitude). Rather the strong connection between Fisher and Shannon can better be explained in terms of Proposition 3.7 that we introduce.

In general, computing the Shannon Information $I(\mathcal{D}; w)$ between a dataset $\mathcal{D}$ and the parameters $w$ of a model is not tractable. However, here we show an example of a simple model that can be trained with SGD, replicates some of the aspects typical of the loss landscape of DNNs, and for which both Shannon and Fisher Information can be estimated easily. According to our predictions in Section 3.3, Figure 1 shows that (center) increasing the temperature of SGD, for example by reducing the batch sizes, reduces both the Fisher and the Shannon Information of the weights (Proposition 3.7), and (left) this is due to the solution discovered by SGD concentrating in areas of low Fisher Information of the loss landscape when the temperature is increased (Proposition 3.6).

The toy model is implemented as follow: The dataset $\mathcal{D} = \{x_i\}_{i=1}^{N}$, with $N = 100$, is generated by sampling a mean $\mu \sim \mathrm{Unif}[-1, 1]$ and sampling $x_i \sim N(\mu, 1)$. The task is to regress the mean of the dataset by minimizing the loss $L_{\mathcal{D}} = \frac{1}{N} \sum_{i=1}^{N}(x_i - \phi(\theta))^2$, where $\theta$ are the model parameters (weights) and $\phi$ is some fixed parametrization. To simulate the over-parametrization and complex loss landscape of DNN, we pick $\phi(\theta)$ as in Figure 1 (right). Notice in particular that multiple value of $\theta$ will give the same $\phi(\theta)$: This ensures that the loss function has many equivalent minima. However, these minima will have different sharpness, and hence Fisher Information, due to $\phi(\theta)$ being more sharp near the origin. Proposition 3.6 suggests that SGD is more likely to converge to those minima with low Fisher Information, which is confirmed in Figure 1 (left), which shows the marginal end point over all datasets $\mathcal{D}$ and SGD trainings. Having found the marginal $q(w)$ over all training and datasets, we can compute the the Shannon Information $I(\mathcal{D}; \theta) = \mathbb{E}_{\mathcal{D}}[\mathrm{KL}(q(w|\mathcal{D}) \| q(w))]$. Note that we take $q(w|\mathcal{D}) = N(w^*, F^{-1})$, where $w^*$ is the minimum recovered by SGD at the end of training and $F^{-1}$ is minimum variance of the estimation, which is given by the Cramér-Rao bound. The (log-determinant of) Fisher-Information can instead easily be computed in close form given the loss function $L_{\mathcal{D}}$. In Figure 1 (center) we show how these quantities change as the batch size $B$ varies. Notice that when $B = N = 100$, the algorithm reduces to standard gradient descent, which maintains the largest information in the weights.

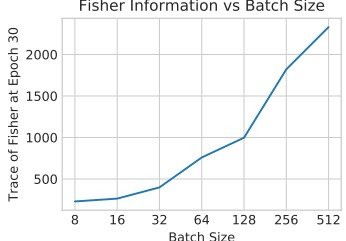 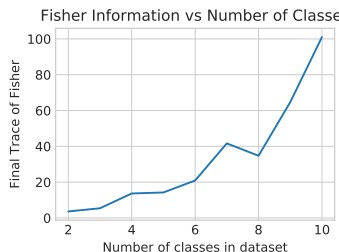

Figure 2: **(Left)** Trace of the Fisher Information of the weights of a ResNet-18 after 30 epochs of training on CIFAR-10, for different values of the batch size. Smaller batch size have smaller Fisher Information, supporting Proposition 3.6. **(Right)** Fisher Information at the end of the training on a subset of CIFAR-10 with only the first $k$ classes: After training on fewer classes, the network has less information in the weights, suggesting that the Information in the Weight has a semantic role.

The value of the Fisher Information cannot be directly compared to the Shannon Information, since it is defined modulo an additive constant due to the improper prior. However, using a proper Gaussian prior that leads to the lowest expected value, we obtain a value of the "Gaussian" Information in the Weights between 4000-5000 nats, versus the $\sim 4$ nats of the Shannon Information: minimizing a much larger (Fisher) bound SGD can still implicitly minimize the optimal Shannon bound.

**Fisher Information for CIFAR-10**   In this section we show the trade-off between amount of information in the weights, complexity of the task (Figure 2, right), and value of $\beta$ (Figure 2, left) on a more realistic problem. More precisely, we validate our predictions on a larger scale off-the-shelf ResNet-18 trained on CIFAR-10 with SGD (with momentum 0.9, weight decay 0.0005, annealing the learning rate by 0.97 per epoch).

First, we compute the Fisher Information (more precisely its trace) at the end of the training training for different values of the batch size (and hence of the "temperature of SGD"). In accordance with Proposition 3.6, Figure 2 (left) shows that after 30 epochs of training the networks with low batch size have a much lower Fisher Information. Second, to check whether the Fisher Information correlates with the amount of information contained in the dataset, we train using only 2, 3, 4, and so on classes of CIFAR-10. Intuitively, the dataset with only 2 classes should contain less information than the dataset with 10 classes, and correspondingly the Fisher Information in the Weights of the network should be smaller. We confirm this prediction in Figure 2 (right).

**Fisher Information and dynamics of feature learning**   We now investigate how the amount of information in the weights of a deep neural network changes during training, in particular to see whether changes in the Fisher Information correspond to the network learning features of increasing complexity. In Figure 3 we train a 3-layer fully connected network on a simple classification problem of 2D points and plot both the Fisher and the classification boundaries during training. Since the network is relatively small, in this experiment we compute the Fisher Matrix exactly using the definition. As different features are learned, we observe corresponding "bumps" in the Fisher information matrix. This is compatible with the hypothesis advanced by Achille et al. (2019), whereby feature learning may correspond to crossing of narrow bottlenecks (high curvature; high Fisher) in the loss landscape, which is followed by a compression phase as the network moves away toward flatter area of the loss landscape.

# B   PROOFS

***Proof of Proposition 3.3.*** For a fixed training algorithm $A : \mathcal{D} \mapsto Q(w|\mathcal{D})$, we want to find the prior $P^*(w)$ that minimizes the expected complexity of the data:

$$P^*(w) = \operatorname*{argmin}_{P(w)} \mathbb{E}_{\mathcal{D}}[C(\mathcal{D})]$$

$$= \operatorname*{argmin}_{P(w)} \Big[ \mathbb{E}_{\mathcal{D}}[L_{\mathcal{D}}(p_w(y|x))] + \mathbb{E}_{\mathcal{D}}[\mathrm{KL}(\, Q(w|\mathcal{D}) \,\|\, P(w) \,)] \Big]$$

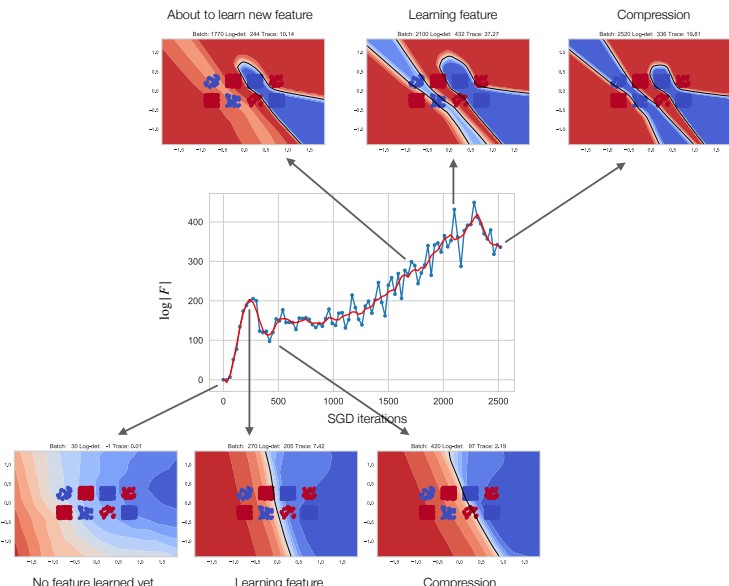

Figure 3: Plot of the log-determinant of the Fisher Information Matrix during training of a 3-layers fully connected network on a simple 2D binary classification task. As the network learns an increasingly complex classification boundary the Fisher increases. Moreover, learning of a new feature correspond to small bumps in the Fisher plot, supporting the idea that feature learning may coincide with crossing of narrow bottlenecks in the loss landscape.

Notice that only the second term depends on $P(w)$. Let $Q(w) = \mathbb{E}_{\mathcal{D}}[Q(w|\mathcal{D})]$ be the marginal distribution of $w$, averaged over all possible training datasets. We have

$$\mathbb{E}_{\mathcal{D}}[\mathrm{KL}(\,Q(w|\mathcal{D}) \,\|\, P(w)\,)] = \mathbb{E}_{\mathcal{D}}[\mathrm{KL}(\,Q(w|\mathcal{D}) \,\|\, Q(w)\,)] + \mathbb{E}_{\mathcal{D}}[\mathrm{KL}(\,Q(w) \,\|\, P(w)\,)].$$

Since the KL divergence is always positive, the optimal "adapted" prior is given by $P^*(w) = Q(w)$, i.e. the marginal distribution of $w$ over all datasets. Finally, by definition of Shannon's mutual information, we get

$$I(w; \mathcal{D}) = \mathrm{KL}(\,Q(w|\mathcal{D})\,\pi(\mathcal{D}) \,\|\, Q(w)\,\pi(\mathcal{D})\,) = \mathbb{E}_{\mathcal{D} \sim \pi(\mathcal{D})}[\mathrm{KL}(\,Q(w|\mathcal{D}) \,\|\, Q(w)\,)]. \qquad \square$$

***Proof of Proposition 3.4.*** Since both $P(w)$ and $Q(w|\mathcal{D})$ are Gaussian distributions, the KL divergence can be written as

$$\mathrm{KL}(\,Q(w|\mathcal{D}) \,\|\, P(w)\,) = \frac{1}{2}\left[\frac{\|\mu\|^2}{\lambda^2} + \frac{1}{\lambda^2}\mathrm{tr}(\Sigma) + k\log\lambda^2 - \log|\Sigma| - k\right],$$

where $k$ is the number of components of $w$.

Let $w^*$ be a local minimum of the cross-entropy loss $L_{\mathcal{D}}(p_w(y|x))$, and let $H$ be the Hessian of $L_{\mathcal{D}}(p_w(y|x))$ in $w^*$. Set $\mu = w^*$. Assuming that a quadratic approximation holds in a sufficiently large neighborhood, we obtain

$$C_\beta(\mathcal{D}; P, Q) = L_{\mathcal{D}}(p_{w^*}(y|x)) + \mathrm{tr}(H \cdot \Sigma) + \frac{\beta}{2}\left[\frac{\|w^*\|^2}{\lambda^2} + \frac{1}{\lambda^2}\mathrm{tr}(\Sigma) + k\log\lambda^2 - \log|\Sigma| - k\right].$$

The gradient with respect to $\Sigma$ is

$$\frac{\partial C_\beta(\mathcal{D}; P, Q)}{\partial \Sigma} = \left[H + \frac{\beta}{2\lambda^2}I - \frac{\beta}{2}\Sigma^{-1}\right]^\top.$$

Setting it to zero, we obtain the minimizer $\Sigma^* = \frac{\beta}{2}(H + \frac{\beta}{2\lambda^2}I)^{-1}$.

Recall that the Hessian of the cross-entropy loss coincides with the Fisher information matrix $F$ at $w^*$, because $w^*$ is a critical point (Martens, 2014). Since $L_{\mathcal{D}}(p_w(y|x))$, and hence $H$, is not normalized by the number of samples $N$, the exact relation is $H = N \cdot F$. Taking the limit for $\lambda \to \infty$, we obtain the desired result. □

***Proof of Proposition 3.7***. It is shown in Brunel & Nadal (1998) that, given two random variables $x$ and $y$, and assuming that $p(x|y)$ is concentrated around its MAP, then the following approximation holds:

$$I(x; y) \approx H(x) - \mathbb{E}_x \Big[ \frac{1}{2} \log \Big( \frac{(2\pi e)^k}{|F_{y|x}|} \Big) \Big], \tag{11}$$

where $F_{y|x} = \mathbb{E}_{y \sim p(y|x)}[-\nabla_x^2 \log p(y|x)]$ is the Fisher Information that $x$ has about $y$, and $k = \dim x$. We want to apply this approximation to $I(w; \mathcal{D})$, using the distribution $p(w|\mathcal{D}) = N(w^*(\mathcal{D}), F(w^*(\mathcal{D}))^{-1})$. Hence, we need to compute the Fisher Information $F_{w|\mathcal{D}}$ that the dataset has about the weights. Recall that, for a normal distribution $N(\mu(\theta), \Sigma(\theta))$, the Fisher Information is given by

$$F_{m,n} = \partial_{\theta_m} \mu^t \Sigma^{-1} \partial_{\theta_n} \mu + \frac{1}{2} \mathrm{tr} \left( \Sigma^{-1} (\partial_{\theta_m} \Sigma) \Sigma^{-1} (\partial_{\theta_n} \Sigma) \right).$$

Using this expression in our case, and noticing that by our assumptions we can ignore the second part, we obtain:

$$F_{w|\mathcal{D}} = \nabla_{\mathcal{D}} w^{*t} F_w(w^*) \nabla_{\mathcal{D}} w^*,$$

which we can insert in eq. (11) to obtain:

$$I(w; \mathcal{D}) \approx H(\mathcal{D}) - \mathbb{E}_{\mathcal{D}} \Big[ \frac{1}{2} \log \Big( \frac{(2\pi e)^k}{|F_{w|\mathcal{D}}|} \Big) \Big]$$

$$= H(\mathcal{D}) - \mathbb{E}_{\mathcal{D}} \Big[ \frac{1}{2} \log \Big( \frac{(2\pi e)^k}{|\nabla_{\mathcal{D}} w^{*t} F_w(w^*) \nabla_{\mathcal{D}} w^*|} \Big) \Big]. \quad \square$$

***Proof of Proposition 4.2***. (1) We need to compute the Fisher Information between $z_n$ and $x$, that is:

$$F_{z|x} = \mathbb{E}_{z \sim p(z|x)} \Big[ - \nabla_x^2 \log p(z = f_w(x)|x) \Big].$$

In the limit of small $\beta$, and hence small $n$, we expand $z_n = f_{w+n}(x)$ to the first-order about $w$ as follows:

$$z_n = f_{w+n}(x) + J_f \cdot n + o(\|n\|)$$

where $J_f$ is the Jacobian of $f_w(x)$ seen as a function of $w$, with $\dim(J_f) = \dim z \times \dim w$. Hence, given that $n \sim N(0, \Sigma_w^*)$, we obtain that $z$ given $x$ approximately follows the distribution $p(z_n|x) \sim N(f_w(x), J_f \Sigma_w^* F_f^t)$.

We can now plug this into the expression for $F_{z|x}$ and compute:

$$F_{z|x} = \mathbb{E}_{z \sim p(z|x)} \Big[ - \nabla_x^2 \log p(z_n|x) \Big]$$

$$= \frac{1}{2} \mathbb{E}_{z \sim p(z|x)} \Big[ \nabla_x^2 \big[ (z - m(x))^t \Sigma(x)^{-1} (z - m(x)) + \log |\Sigma(x)| \big] \Big]$$

$$= \nabla_x f_w \cdot \Sigma_w^{-1} \nabla_x f_w.$$

(2) We now proceed to estimate the Shannon mutual information $I(z; x)$ between activations and inputs. In general, this does not have a closed form solution, rather we use again the approximation of Brunel & Nadal (1998) as done in the proof of Proposition 3.7. Doing so we obtain:

$$I_{\mathrm{eff}}(x; z) \approx H(x) - \mathbb{E}_x \Big[ \frac{1}{2} \log \Big( \frac{(2\pi e)^k}{|F_{z|x}|} \Big) \Big]$$

$$\approx H(x) - \mathbb{E}_x \Big[ \frac{1}{2} \log \Big( \frac{(2\pi e T)^k}{|\nabla_x f_w \cdot J_f F_{\mathbf{w}} J_f^t \nabla_x f_w|} \Big) \Big]. \quad \square$$

