# OpenReview forum: "Where is the Information in a Deep Network?"
_ICLR.cc/2020/Conference — Reject_

### Official Review · AnonReviewer3 · 2019-10-18
**Official Blind Review #3**

**Rating:** 6

**Review:**

This paper presents a theoretical account of information encoded within deep neural networks subject to information theoretic measures. In contrast to other efforts that examine information encoded in weights, this work emphasizes the effective information in the activations. This characterization is further related to information in the weights, and a theoretical justification is made for what this means with respect to properties of generalization and invariance in the network.
The notion of attaching the weights that represent the training set to the activations that accord with the test set in a theoretical framework is interesting. In practice, I would have liked to see a bit more attachment of the theoretical formalisms to the empirical justification that follows the references. This, however, is a matter of personal bias as I don't typically produce papers that are principally theoretical contributions in my own work. Overall, the content of the paper seems sound and the theoretical and empirical justifications seem well founded but I also can't claim to be an expert in this area.

**Experience Assessment:**

I do not know much about this area.

**Review Assessment: Checking Correctness Of Derivations And Theory:**

I assessed the sensibility of the derivations and theory.

**Review Assessment: Checking Correctness Of Experiments:**

I assessed the sensibility of the experiments.

**Review Assessment: Thoroughness In Paper Reading:**

I read the paper at least twice and used my best judgement in assessing the paper.

---

> ### Author Response · Authors · 2019-11-15
> **Response to Reviewer 3**
>
> We thank the reviewer for the comments and suggestions.
>
> >>  I would have liked to see a bit more attachment of the theoretical formalisms to the empirical justification that follows the references.
>
> We have now updated the appendix to better introduce each experiment, to hopefully better explain the connections with both theory we develop and to the previous literature.

---

### Official Review · AnonReviewer1 · 2019-10-24
**Official Blind Review #1**

**Rating:** 8

**Review:**

The paper deals with where the information is in a deep network and how information is propagated when new data points are observed. The authors measure information in the weights of a DNN as the trade-off between network accuracy and weight complexity. They bring out the relationships between Shannon MI and Fisher Information and the connections to PAC-Bayes bound and invariance. The main result is that models of low information generalize better and are invariance-tolerant.

The paper is very well written and concepts are theoretically-well documented.

In Definition 3.1 for the ‘Information in the Weights’, how does the complexity of the task vary with \beta? Is the Pareto curve provided in the paper?

**Experience Assessment:**

I do not know much about this area.

**Review Assessment: Checking Correctness Of Derivations And Theory:**

I assessed the sensibility of the derivations and theory.

**Review Assessment: Checking Correctness Of Experiments:**

N/A

**Review Assessment: Thoroughness In Paper Reading:**

I read the paper at least twice and used my best judgement in assessing the paper.

---

> ### Author Response · Authors · 2019-11-15
> **Response to Reviewer 1**
>
> We thank the reviewer for the positive comments.
>
> >> In Definition 3.1 for the ‘Information in the Weights’, how does the complexity of the task vary with $\beta$? Is the Pareto curve provided in the paper?
>
> This is partially answered empirically in Fig. 2. (left), where we keep the task constant but we change the batch size of SGD, which amounts to reducing beta by Proposition 3.7. This results in the expected increase in the Fisher Information of the weights. In (Right) we keep beta constant, but we increase the complexity of the task by adding more classes. This also results in an increase in the Fisher Information.

---

### Official Review · AnonReviewer2 · 2019-10-28
**Official Blind Review #2**

**Rating:** 6

**Review:**

Summary of the paper:
This is a theoretical paper that builds on top of Achille and Soatto (2018), Achille et al. (2019), McAllester (2013), and Berglund (2011),. The paper attempts to answer the relationship between the inductive bias of SGD, generalization of DNNs, and the invariance of learned representation from an information theoretical point of view.
The paper mentioned many interesting links. In my opinion, the contributions are the following:
1. Invoking the theoretical result of Berglund (2011) to justify why Fisher information is relevant -- SGD tends to avoid local minima with high Fisher information.
2. Relating the Fisher information and the stability of SGD to I(w; D).
3. Introducing the definition of effective information in the activation, and show that which is closely related to the Fisher information.

About the rating:
This is basically a good paper but I have a few concerns:
1. A large fraction of this paper are taken from Achille and Soatto (2018), Achille et al. (2019).
2. In terms of impact, the paper is somehow incomplete -- it only demonstrates that the Fisher information is important, but the insights didn't lead to any substantial improvement over the current deep learning framework.

Detailed comments:
1. In my opinion, defining "information in the weights for the task D" by KL(Q||P) is inaccurate.
The weights themselves are information, which form a representation or a lossy compression of the data (which is also an information).
According to the rate-distortion theory, what we care about is the amount of information the representation attains rather than "where" the information are. Therefore, we should talk about "rate" or "amount of information" or "mutual information" rather than "information" itself.
A missing reference regarding this point:
Hu et al. "β-BNN: A Rate-Distortion Perspective on Bayesian Neural Networks." 2018,
which derives the information Lagrangian directly from rate-distortion theory.
2. More discussions on Xu and Raginsky (2017) is expected, since it proposes to use I(w; D) as a generalization bound. It seems, in terms of generalization, minimizing I(w; D) is a sufficient condition while minimizing Fisher is a necessary condition.
3. There are in fact 4 key aspects: sufficiency, minimality, invariance and generalization. It would be great to have a theorem to summarize the relationships between them.
4. Could you elaborate on the footnote 3?

**Experience Assessment:**

I have published one or two papers in this area.

**Review Assessment: Checking Correctness Of Derivations And Theory:**

I assessed the sensibility of the derivations and theory.

**Review Assessment: Checking Correctness Of Experiments:**

I assessed the sensibility of the experiments.

**Review Assessment: Thoroughness In Paper Reading:**

I read the paper at least twice and used my best judgement in assessing the paper.

---

> ### Author Response · Authors · 2019-11-15
> **Response to Reviewer 2 (part 1)**
>
> We thank the reviewer for the many thoughtful suggestions. We reply to each point in order:
>
> Concerning relations with Achille and Soatto (JMLR 2018), that paper uses Shannon's framework and effectively considers the weights as stochastic, thus not addressing the computability of information for deterministic maps, where it is often degenerate.  One interpretation of our work is to reconcile that paper with the work criticizing the use of the Information Bottleneck for deterministic networks. This requires formally connecting  the Shannon Mutual Information of the weights to the Fisher (Proposition 3.7, which we also verify empirically in the appendix, and which replaces the much looser bound using the curvature suggested by Achille and Soatto (2018)) and to introduce the notion of effective mutual information of the activations, which we also connect to the Fisher Information (Proposition 4.2). Second, our aim is to formally connect the dynamics with SGD (in terms of both stability and flatness of the solution found) with the information in a DNN. This is not done in any of the references cited.
>
> Concerning relations with Achille et al. (2019), as we say in the opening of Sect. 3, Sections 3.1 and 3.2 are derived from that preprint and included for completeness. The main results of our paper are in Sect. 3.3 and 4, whereas Achille at al. (2019) focus on defining a distance between learning tasks, which we do not address here.
>
> In terms of impact, indeed, our aim was to obtain clarity around the notion of information, both in the the weights and activations of a DNN, that has caused some confusion in the literature and occasionally contradictory or (accidentally) misleading claims. We also introduce  stronger connections between information and the optimization dynamics of a DNN. This, we believe, helps paint a more complete picture of the current landscape of information in Deep Learning. We did not set out to improve current deep learning frameworks, but we hope this work will help us and others at least understand how different fundamental quantities are related in DNN, when a model can be expected to "work," hopefully quantify how "well" it works, and to relate this to the complexity of the learning task, which is not often formalized in deep learning.
>
> Regarding the detailed comments:
>
> >> we should talk about "rate" or "amount of information" or "mutual information" rather than "information" itself
>
> This is a good point, and we have updated the definition to reflect this. We originally tried to avoid naming it "mutual information" or "rate" to make it clear that the notion is valid even if the dataset is not considered a random variable (like it would in rate-distortion theory), but we are glad to change it to the less ambiguous "amount of information".

---

> > ### Author Response · Authors · 2019-11-15
> > **Response to Reviewer 2 (part 2)**
> >
> > >> A missing reference regarding this point:  Hu et al. "β-BNN: A Rate-Distortion Perspective on Bayesian Neural Networks." 2018
> >
> > We thank for the reference to the workshop paper, which we were not aware of and we will reference. We should notice that while Hu et al. (2018) derive their framework as an approximation of mutual information, which  assumes a distribution over both the datasets and the weights, we derive our notion of information (or indeed "amount of information'') for one particular given dataset and set of weights.
> >
> > >> 2. More discussions on Xu and Raginsky (2017) is expected, since it proposes to use I(w; D) as a generalization bound.
> >
> > We updated the paper to discuss the work more at length. Specifically, Xu and Raginsky prove a bound on generalization using the "information" stability of the algorithm. We show that, in realistic settings for a DNN, if the optimization algorithm is "stable" (in the sense that the final point of the optimization does not change much for perturbations of the dataset), then, together with the minimization of the Fisher, this implies "information" stability (and hence generalization, by either the PAC-Bayes bound, which we use, or the bound proposed by Zu and Raginsky (2017) and related works).
> >
> > The relaxed loss they propose is related to our loss. However, they propose the Gibbs algorithm to minimize the loss, which is not practical for a DNN. On the other hand, we show that the more practical SGD algorithm approximately minimizes a term of that form (the free energy in Proposition 3.6), thus connecting theory with common practice and emphasizing the role of the dynamics of the training process in order to get good generalization, which is not studied by Zu and Raginsky (2017).
> >
> > Moreover, the success of deep learning hinges on the fact that once trained on a (large) dataset, the representation can be used on a new dataset. This is not captured by the bound proposed by Zu and Raginsky (2017). By connecting the information in the weights with information in the activations, we get some guarantees to the invariances learned by the representation that are going to be transfered to the new dataset.
> >
> > >> It seems, in terms of generalization, minimizing I(w; D) is a sufficient condition while minimizing Fisher is a necessary condition.
> >
> > Regarding the relationship between I(w; D) and the Fisher, the exact relationship depends on the algorithm. It could be that the algorithm always picks the same point in weight space with a high Fisher, regardless of the task (notice that the Fisher depends only on the point and the input distribution, and not on the task labels). This minimizes I(w; D) since w = A(D) is constant, but maximizes the Fisher. (This example does not, of course, satisfy the hypotheses of our Prop. 3.7 as p(D|w) is uniform, rather than concentrated on a single dataset).
> >
> > >> 3. There are in fact 4 key aspects: sufficiency, minimality, invariance and generalization. It would be great to have a theorem to summarize the relationships between them.
> >
> > That's a great suggestion, we will add the theorem as summary in the discussion in the camera-ready.
> >
> > >> 4. Could you elaborate on the footnote 3?
> >
> > The core idea of Proposition 3.7 is to measure how perturbations of the dataset D affect the minimum, also in relation to the amount of noise in SGD, which in turn is proportional to the Fisher. If, for example, changing one single label of the dataset slightly shifts the convergence point by some amount which is neglegible with respect to the noise, then the weights are not carrying much information about that sample. Proposition 3.7 formalizes this notion; however, it is easier to derive it while considering continuous perturbations of the dataset, rather than discrete ones. One could, for example, consider $\mathcal{D}_\theta = \{(x_i, f_\theta(x_i))\}_{i=1}^N$, where the label (which we assume being a soft label) is parametrized by a function $f_\theta$. Perturbing theta now changes the label in a continuous way. If $f_\theta$ is an expressive-enough family of functions (e.g., a DNN itself, in which case this would be similar to a teacher-student setting), then any dataset on a fixed domain can be expressed in this way.
> >
> > An alternative way to explain this would be to assume that we have a fixed pool of data points, and that $\mathcal{D}$ is constructed by sampling those data points with some categorical probability distribution theta. Changing $\theta$ will now sample different datasets. Using, for example, the Gumbel-max trick, the final dataset can be considered a differentiable stochastic function of theta.

---

### Comment · Area_Chair1 · 2019-11-15
**A few concerns**

There seems to be only one expert reviewer on this paper, and so I've gone ahead and read the paper carefully. I have a few questions / concerns. I understand these are coming rather late, but I will make sure that I get to hear your responses.

1. Isotropic noise assumption

I have a concern about the isotropic assumption made of the minibatch noise, allowing the authors to link results about diffusions to the behavior of SGD. Indeed, the minibatch noise of SGD is in fact definitely not isotropic: you would expect SGD to actually settle into any minimum where it reaches zero error, and that is indeed what we see in practice in many vision problems. This problem alone seems to be rather problematic for many of the claims that build on this connection. I would argue that your results seem to be about Langevin dynamics, not SGD.

2. Curvature assumption in Proof of Prop 3.4

After the remark following the proof of Prop 3.4, there is a statement: "Note that there is no assumption that the curvature of the loss be constant near convergence." However, inspecting the proof on page 14, I see the statement "Assuming that a quadratic approximation holds in a sufficiently large neighborhood, ...". Isn't that precisely a constant-curvature assumption? Can you elaborate on both statements and also discuss the relationship, if any?

3. Informal "summary" of Prop 3.6

After Prop 3.6, the authors seem to suggest that the escape time result implies something directly about the long-run behavior of the Markov chain: "We can informally summarize the above statement as saying that SGD, rather than minimizing directly the loss function, minimizes a free energy". Can you provide proof of any connection between the escape time and the long run behavior? Issue #1 also bears on this connection, since SGD does not have isotropic noise: this result is not about SGD.

3. Information in the weights under uninformative prior

In Prop 3.4, the limit as lambda diverges is taken. This will generally take the KL divergence to infinity as well. This divergence is caused by the k/2 log lambda^2 term in Equation 7. The authors claim that this term does not depend on Q and thus can be ignored. However, Theorem 3.2 and subsequent claims about generalization, rely on the entire "Information in the Weights" quantity. One cannot simply discard a term that is causing the bound to race off to infinity without some careful argument. I don't see any such argument. Can you argue why later claims about generalization are meaningful despite this divergence of the bound?

4. What is Q?

Having read the paper, I'm somewhat confused what the post-distribution Q is meant to be in this story. Is Q the distribution of the weights produced by SGD?

5. Relationship with Xu-Raginsky.

The authors seem to want to distance themselves from the Xu-Raginsky (and subsequent Pensia et al results). I don't think this is possible. First, the authors are pointing at PAC-Bayes bounds (which are tail bounds) but then arguing through the expectation of the bound (in order to get mutual informations), and so one eventually arrives at bounds in expectation. Bounds in expectation are precisely what Xu and Raginsky provide. There is a tight connection between Xu and Raginsky and PAC Bayes bounds as well: they are both derived from Donsker Varadhan.

---

> ### Author Response · Authors · 2019-11-15
> **Response to the Area Chair**
>
> We thank the Area Chair for the questions, which give us an opportunity to clarify possible misunderstandings. We give detailed responses to the points raised below, that reaffirm that the claims made are correct, but will add further color to the discussion, clarify the nomenclature for SGD and amend the statement of Prop. 3.4 to clear any possible confusion.
>
> >> 1. Isotropic noise assumption
>
> The noise of SGD being non-isotropic is a known fact, as we describe at the end of page 3, and not a problem for our claims. There are only two claims that reference isotropy, Proposition 3.4, that has however no connection with SGD,  and Proposition 3.6. Specifically, the expectation in the Proposition 3.6 is computed under the assumption of isotropic noise, and the value is only approximate if the assumption, stated clearly in the claim, is not satisfied. The Area Chair is right in that, while the claim is correct, if its assumptions are violated, the resulting approximation may be poor. We will clarify this in the discussion following the theorem, to avert possible confusion.
>
> The computation of the escape time in the anisotropic case is known to be a hard problem. We deem the result in the isotropic case useful nevertheless because, even if the value of the expectation is different in the anisotropic case, the trends are the same. We should also mention that the extension of classic results in Langevin Dynamics to the case of SGD is the subject of active investigation, for instance https://arxiv.org/pdf/1907.03215.pdf.
>
> More importantly, successive claims in our work do not build directly on Proposition 3.6, but rather on the fact that the minimum to which SGD converge is flat. This fact has been verified empirically by multiple independent studies, for different tasks, different architectures, and different variants of SGD. So, even if one were to dismiss Proposition 3.6 as not pertinent to  SGD, all subsequent results hold if one accepts that it converges to low-curvature regions of the loss landscape. Proposition 3.6 shows that, albeit under strong assumptions, a stochastic gradient algorithm run for a finite time is more likely to settle on a flat minimum rather than a sharp one.
>
> Regarding the fact that SGD converges to a minimum with zero error, we think it is more appropriate to say that SGD tends to converge to areas of the loss landscape with very low loss and curvature, which it will not escape in a short time. However, the core idea of Proposition 3.6 -- and indeed the reason why we use it instead of a simpler result on the stationary distribution, which would hold only asymptotically -- is to suggest that SGD can easily escape very sharp minima (or, more generally, any area of the loss with high curvature) in its path before settling. Hence, when we stop the optimization after a finite time, a noisy gradient descent algorithm will be more biased toward converging to flatter areas of the landscape than a corresponding non-noisy gradient flow.
>
> The only other claim where isotropy is mentioned is Proposition 3.4 in the choice of the prior, which is however not related to SGD.
>
> We will add a section in the appendix to elaborate on this issue, add references to asymptotic analyses that would clarify potential confusion, and to discuss in detail the relation between SGD, Langevin Dynamics, and the role of the assumptions on the claims.

---

> > ### Author Response · Authors · 2019-11-15
> > **Response to the Area Chair (part 2)**
> >
> > >> 2. Curvature assumption in Proof of Prop 3.4
> >
> > The fact that the quadratic approximation is valid does not imply that the curvature is constant. It simply means that higher-order terms are negligible, which can happen while the curvature changes along the path.  Proposition 3.4 gives the optimal value of the information up to a second order term. If the curvature is not constant around the minimum, as it is likely to be, the value of the information computed will be wrong by a term which will go to zero as $\beta \to \infty$.
> >
> > We will edit the proof to clarify where the second-order approximation is used, which is simply to estimate the term $\mathbb{E}_{w\sim q(w)}[L_\mathcal{D}(w)]$ around the mean of $q(w)$ using the gradient and Hessian of $L_\mathcal{D}$, and not in any way that requires the loss to be exactly quadratic near the minimum.
> >
> > >> 3. Informal "summary" of Prop 3.6
> >
> > The point of the claim, where we refer to various stochastic optimization methods collectively as ‘SGD’,  is that the stochasticity adds a diffusion/regularization term to the loss, so what is minimized is not the original loss, but a regularized one that has the form of free energy. Connecting the free energy to the long-term behavior and time-to-escape from minima is one of the main objectives of non-equilibrium dynamics and Kramer’s theory, and beyond the scope of a single conference paper. However,  the informal take-away from Prop. 3.6 is that, when sharper and flatter minima with the same loss are connected, a noisy process will tend to escape the sharp minimum towards the flat minimum. This can be interpreted as the process moving in the direction of the minimum with lower free energy, where the free energy accounts for both the loss and the curvature. This argument could be made formal, but at the cost of readability. We prefer an informal summary, leaving to other papers (including some of those cited) to prove more formally.
> >
> > >>  3. Information in the weights under uninformative prior
> >
> > Indeed, point well taken, and we will clarify. Specifically, we do not intend to claim that,for $\lambda \to \infty$, eq. 7 provides a valid generalization bound. Rather, the correct (finite) expression, for any finite choice $\lambda$, is given by:
> > $$KL(p||q) = \frac{\|{w^*}\|^2}{\lambda^2}+ \frac{1}{\lambda^2} \operatorname{tr}(\Sigma) + k \log \lambda^2 - \log |\Sigma|-k + o(1), \quad(*)$$
> > where $\Sigma=\frac{\beta}{2} (F + \frac{\beta}{2\lambda^2} I)^{-1}$. The proof is identical to that in the appendix, simply without taking the limit wrt. lambda. Note that $F$ still plays the same role in this expression, which is just slightly more cluttered. What we meant to say by ignoring the constant term is that, if some variational optimization algorithm aims to minimize the  KL term through a gradient descent process, the gradient it will receive will (under mild regularity assumptions) be well defined in the limit $\lambda \to \infty$.
> >
> > Admittedly, our choice notation and presentation aimed at simplicity was made at the expense of clarity, so we will rewrite the proposition using the expression (*), and change the statement accordingly.

---

> > > ### Author Response · Authors · 2019-11-15
> > > **Response to the Area Chair (part 3)**
> > >
> > > >> 4. What is Q? Having read the paper, I'm somewhat confused what the post-distribution Q is meant to be in this story. Is Q the distribution of the weights produced by SGD?
> > >
> > > It is not. As we describe after Definition 3.1, Q is an arbitrary choice, corresponding to an encoding for the weights, irrespective of how the weights are obtained (in particular, throughout most of the paper, the weights are not assumed to be a random variable, but rather a fixed vector). To add color to our description in Sect. 3, informally, we are interested in how much information we need to encode the (fixed) weights of the network. Since the weights are a continuous vector, encoding them exactly would require an infinite amount of information (if using a continuous prior P, such as a Gaussian, for the encoding). However, encoding the weights exactly is pointless, as we know that it is possible to perturb the weights (which is often referred to as “adding noise’’ even if there is no stochastic process at play) without substantially increasing the risk. On the other hand, encoding “noisy weights” can be done with a finite amount of information (for example, one could discretize the weights using the standard deviation of the noise along each parameter as part of the quantization process). For this reason, it is natural to think of the distribution Q as the “amount of noise that could be added to the weights” (even if no noise is actually added to the weights), while not increasing the loss by more than a fixed amount; then, KL(Q||P) is the coding length for that particular set of weights, using the particular (arbitrary) choice of code specified by P and Q, when allowing a lossy compression with this noise. We did not belabor this point since a more formal argument to this effect has already been given by Hinton and Van Camp (1993), as we indicate in Sect. 1 and Sec. 3.  However, we concur that the role of Q can be confusing, so we will add a discussion in the appendix to clarify.
> > >
> > > >> 5. Relationship with Xu-Raginsky.
> > >
> > > We will revise the narrative to make sure we do not give the impression that we want to distance our results from the work of Xu and Raginsky, since this is not out intention. On the contrary, we are quite intrigued by the connections, which are also described in more detail in our prior answer to reviewers.  We also agree that the PAC-Bayes  and information stability bounds are related. What this work is trying to do is not to introduce new information bounds (we are happy to use either PAC-Bayes or Xu et al.), but rather to show how the learning dynamics affects the information bounds: Not just by decreasing the information contained in each gradient step because of the noise (an argument often exploited in the literature), but also through the geometry of the loss landscape (flat minima) and the “path” stability of the algorithm. Moreover, we want to show that, in a DNN, low information does not solely mean better generalization (as those bounds already show), but also better properties of the learned representation. We do so by introducing the notion of effective information of the activations, which also tries to solve some formal issues with the Information Bottleneck theory for the study of the activations.
> > >
> > > We are happy to further clarify these relationships in the body of the paper, also including the relation between PAC-Bayes and the bound of Xu and Raginsky, and related works, mentioned by the AC.

---

### Decision · Program_Chairs · 2019-12-19

**Decision:**

Reject

**Comment:**

This paper is full of ideas. However, a logical argument is only as strong as its weakest link, and I believe the current paper has some weak links. For example, the attempt to tie the behavior of SGD to free energy minimization relies on unrealistic approximations. Second, the bounds based on limiting flat priors become trivial. The authors in-depth response to my own review was much appreciated, especially given its last minute appearance. Unfortunately, I was not convinced by the arguments. In part, the authors argue that the logical argument they are making is not sensitive to certain issues that I raised, but this only highlights for me that the argument being made is not very precise.  I can imagine a version of this work with sharper claims, built on clearly stated assumptions/conjectures about SGD's dynamics, RATHER THAN being framed as the consequences of clearly inaccurate approximations. The behavior of diffusions can be presented as evidence that the assumptions/conjectures (that cannot be proven at the moment, but which are needed to complete the logical argument) are reasonable. However, I am also not convinced that it is trivial to do this, and so the community must have a chance to review a major revision.

---

> ### Author Response · Authors · 2020-01-20
> **Response to Paper Decision**
>
> We are surprised and disappointed by the Area Chair's decision, and the process used to arrived at it, considering that all three reviews are positive, that we have responded to the Area Chair's own review – itself an anomaly – despite it being posted hours before the closing of the rebuttal period. Simple modifications that we could have made during the rebuttal period – but were not allow to given the timing of the extra review – would had addressed the objections. If the Area Chair truly still has substantive concerns, we invite him/her to reach out to us and we will be delighted to help him/her work through the logical arguments, and the assumptions they are actually based on.
>
> For now, we respond to each comments in-line, which are reflected in the updated version to be posted on ArXiv.
>
> >> A logical argument is only as strong as its weakest link, and I believe the current paper has some weak links.  For example, the attempt to tie the behavior of SGD to free energy minimization relies on unrealistic approximations.
>
> We assume the Area Chair refers to the approximation of the noise in SGD as being isotropic in the proof of Proposition 3.6. Apart from the fact that the anisotropic dynamics of SGD have only been worked out recently for the one-dimensional case, as we have already mentioned in the rebuttal, the actual numbers of the escape rate will surely change, but the statement of the claim, concerning the fact that the probability of escape will be higher for high-curvature wells than for low-curvature, is still true.
>
> >> Second, the bounds based on limiting flat priors become trivial.
>
> We are not sure of what bounds the Area Chair refers to, since we never compute any bounds that are based on limiting flat priors, and indeed argue in the rebuttal that doing so would be nonsensical. We assume the AC means the limit $\lambda \to \infty$ in the expression of the Fisher, to which we respond next.
>
> >> In part, the authors argue that the logical argument they are making is not sensitive to certain issues that I raised, but this only highlights for me that the argument being made is not very precise.
>
> We assume the Area Chair refers to the constant term in the Fisher Information, which becomes infinite as the prior becomes improper. For any constant value of $\lambda$ this term is not present in the gradients or in the difference of the free energies, nor does it have any effect on optimization, which is what matters in the analysis. While $\lambda$ surely affects the numerical value of the Information in the Weights, this is no different than what happens in defining differential entropy as a limit of the KL divergence with an improper uniform prior, which similarly leads to a diverging term which is ignored without drama in many situations.
>
> >>  I can imagine a version of this work with sharper claims, built on clearly stated assumptions/conjectures about SGD's dynamics, RATHER THAN being framed as the consequences of clearly inaccurate approximations.
>
> It appears the sticky point is again the use of isotropic noise in Proposition 3.6. Indeed, we state clearly that we assume isotropic noise, and we clearly acknowledge that SGD noise is not isotropic in deep networks. Knowingly using assumptions that are not satisfied in practice is not uncommon in analyzing real signals (e.g., the band-limited assumption in the classical Sampling Theorem), and is done because it makes the proof possible (as we already pointed out, anisotropic non-asymptotic analysis is in its infancy), and because it highlights underlying mechanisms concerning escape from sharp minima that are manifest regardless of isotropy.
>
> >> The behavior of diffusion can be presented as evidence that the assumptions/conjectures (that cannot be proven at the moment, but which are needed to complete the logical argument) are reasonable.
>
> The logical arguments we present are complete and the statements are falsifiable.  Some readers may see a gap in extending conclusions drawn for stochastic optimization with isotropic noise to the anisotropic case. We do not share such concerns but, regardless, the claims are for the isotropic case, and they are valid with no logical gaps.
>
> >> However, I am also not convinced that it is trivial to do this, and so the community must have a chance to review a major revision.
>
> The changes needed to address these objections are simple, and do not require changing any of the claims or proofs. We do not wish to put weights on the technicalities – for instance computing the exact escape time, which would require assuming anisotropic noise – since this paper is about  properly defining a notion of information in the weights, its relation to optimization, and to the information in the activations. However, others are welcome to conduct the analysis in the anisotropic case, which is well beyond the scope of our paper. Doing so would take the understanding of information in deep networks one further step forward.